# Monte Carlo Dropout for Uncertainty Estimation and Motor Imagery Classification

**DOI:** 10.3390/s21217241

**Published:** 2021-10-30

**Authors:** Daily Milanés-Hermosilla, Rafael Trujillo Codorniú, René López-Baracaldo, Roberto Sagaró-Zamora, Denis Delisle-Rodriguez, John Jairo Villarejo-Mayor, José Ricardo Núñez-Álvarez

**Affiliations:** 1Department of Automatic Engineering, Universidad de Oriente, Santiago de Cuba 90500, Cuba; daily@uo.edu.cu; 2Serconi, Holguín 80100, Cuba; rtrujillo@uo.edu.cu; 3Zimtronic, Miami, FL 33222, USA; rlopez@zimtronic.com; 4Department of Mechanical Engineering, Universidad de Oriente, Santiago de Cuba 90500, Cuba; sagaro2001@gmail.com; 5Postgraduate Program in Electrical Engineering, Federal University of Espirito Santo, Vitoria 29075-910, Brazil; delisle05@gmail.com; 6Department of Physical Education, Federal University of Paraná, Curitiba 80050-520, Brazil; jvimayor@gmail.com; 7Department of Energy, Universidad de la Costa, Barranquilla 080002, Colombia

**Keywords:** Brain–Computer Interfaces, Monte Carlo dropout, motor imagery, Shallow Convolutional Neural Network, uncertainty estimation

## Abstract

Motor Imagery (MI)-based Brain–Computer Interfaces (BCIs) have been widely used as an alternative communication channel to patients with severe motor disabilities, achieving high classification accuracy through machine learning techniques. Recently, deep learning techniques have spotlighted the state-of-the-art of MI-based BCIs. These techniques still lack strategies to quantify predictive uncertainty and may produce overconfident predictions. In this work, methods to enhance the performance of existing MI-based BCIs are proposed in order to obtain a more reliable system for real application scenarios. First, the Monte Carlo dropout (MCD) method is proposed on MI deep neural models to improve classification and provide uncertainty estimation. This approach was implemented using Shallow Convolutional Neural Network (SCNN-MCD) and with an ensemble model (E-SCNN-MCD). As another contribution, to discriminate MI task predictions of high uncertainty, a threshold approach is introduced and tested for both SCNN-MCD and E-SCNN-MCD approaches. The BCI Competition IV Databases 2a and 2b were used to evaluate the proposed methods for both subject-specific and non-subject-specific strategies, obtaining encouraging results for MI recognition.

## 1. Introduction

Deep neural network (DNN) techniques have gained enormous acceptance in the scientific community with respect to other machine learning techniques. For this reason, DNN is becoming more attractive for various research areas, such as language processing, computer-assisted systems, medical signal processing, and autonomous vehicles, among others. Particularly, Motor Imagery (MI)-based Brain–Computer Interfaces (BCIs) by using DNN have proven potential for MI tasks classification with good discrimination. BCI systems constitute an alternative communication pathway for patients with severe neural impairments, and they consist of brain signal acquisition and processing, which is then translated into control commands for robotic devices, such as exoskeletons, wheelchairs, etc. [1]. Despite the impressive accuracy of employing DNN-based BCIs, these approaches may produce overconfident predictions. Moreover, the analysis to quantify the uncertainty of predictions is still a challenge. Overconfident incorrect predictions may be undesirable; hence, an analysis for uncertainty quantification is crucial to guarantee more robust BCIs with reliable responses, and consequently, making them suitable for real-life scenarios [2,3].

The uncertainty estimation in deep learning is an open area. A variety of research has focused on dealing with the uncertainty of DNN models in some fields, such as medicine [4,5,6], autonomous navigation systems [7,8], robotic [9], and natural language processing [10] to improve the decision making of recognition systems. For medical applications, the uncertainty analysis has been widely used for diseases diagnoses, such as COVID-19 [11], tuberculosis [12], ataxia [13], cancer [14], diabetes [15,16], and epilepsy [17]. Nevertheless, to the best of our knowledge, the uncertainty analysis in MI-based BCIs has not been reported. Dealing with EEG signals that have naturally a poor signal-to-noise ratio, small amplitudes, and high variability intra-and inter-subjects and inter-sessions make the EEG-based BCI systems more difficult and prone to produce unreliable predictions, affecting their effectiveness. In fact, healthy BCI novices and people with attention deficit suffering also with severe motor impairments [18,19] can have problems for appropriately executing MI tasks, which may increase the uncertainty of BCI predictions.

Therefore, we hypothesize that an MI-based BCI including methods for uncertainty analysis may improve the closed-loop between both user and end-effector, enhancing neuroplasticity [20].

Regarding uncertainty estimation, Bayesian neural networks (BNN) [21,22] are a probabilistic version of neural networks, which are intrinsically suitable to estimate uncertainty. The variational inference [22] is commonly used to approximate the posterior model by using simple variational distribution, such as the Gaussian distribution. However, the training process of these networks is more complex, and the trained networks may not always offer superior accuracy results.

Gal and Ghahramani [23] introduced a method for determining the model uncertainty. They noted that training any neural network by using dropouts, typically used for preventing overfitting, can be interpreted as an approximate inference of the weight’s posterior. In short, applying dropout at test time, this method makes multiple forward passes with the trained model. Then, for a given input, the prediction and the model uncertainty can be statistically estimated. This popular method, known as Monte Carlo dropout (MCD) [13,16,23,24,25,26], has become attractive in practice, since it scales well to large amounts of data, and it does not require the change of existing model architectures. MCD can be interpreted as an averaging ensemble of many networks with shared weights [24,27,28].

The objective of this study is to propose methods for accuracy improvement and uncertainty analysis in a deep neural network scheme for MI-based BCIs in order to obtain robust predictions, and therefore, more reliable responses of robotic devices during motor rehabilitation. An existing Shallow Convolutional Neural Network (SCNN) [29] scheme was used in our study to test the main contributions of this paper, such as SCNN combined with Monte Carlo dropout (SCNN-MCD) to classify motor imagery EEG in BCI applications and estimate the model uncertainty. We investigate with SCNN-MCD how different uncertainty measures correlated with the predictive accuracy and also introduce a threshold method that rejects EEG inputs that produce predictions of very high uncertainty in cases when we should not rely on the prediction and minimizes the error rate of the classifier. The SCNN-MCD model was tested on the BCI Competition IV datasets 2a and 2b, and it yields significant accuracy improvement compared to the state-of-the-art. Finally, the MCD of deep ensemble models is explored and evaluated on both datasets, also obtaining promising results.

## 2. Materials and Methods

### 2.1. Databases Description

Two popular public datasets from BCI Competition IV, specifically datasets 2a and 2b [29,30,31,32,33,34,35,36,37], were used.

Dataset 2a: This dataset contains EEG signals of different MI tasks (left hand, right hand, tongue, and foot), which were recorded from nine healthy subjects on 22 locations (Fz, FC1, FC3, FCz, FC2, FC4, C5, C3, C1, Cz, C2, C4, C6, CP3, CP1, CPz, CP2, CP4, P1, Pz, P2, and POz) with a sampling rate at 250 Hz. A total of two sessions on different days was performed for each subject, completing a total of 288 trials per session. The first session was used here for training, and the other session was employed for testing, as done in previous works [29,30,31,32,33,34].

Dataset 2b: This dataset comprises EEG signals from two MI tasks (left hand and right hand), which were collected from nine subjects over three bipolar EEG channels (around C3, Cz, and C4) with a sampling rate at 250 Hz. Each subject completed a total of five sessions summing 720 trials (first two sessions with 120 trials each, and the last three of 160 trials each). Then, the first three sessions were used for training, while the last two sessions were used for testing, as done in previous works [29,35,36,37].

### 2.2. Preprocessing

Some studies [38,39,40,41,42,43] reported that real or imagined unilateral movements can attenuate or enhance the amplitude of mu (from 8 to 12 Hz) and beta (from 13 to 30 Hz) EEG rhythms over the primary motor cortex in both contralateral and ipsilateral hemispheres, respectively, which are phenomena known as event-related desynchronization/synchronization (ERD/ERS). For this reason, as shown in Figure 1, the EEG signals are band-pass filtered here in a frequency range from 4 to 38 Hz through a Butterworth filter, aiming to preserve the ERD and ERS rhythms, rejecting undesirable physiological and non-physiological artifacts. Next, each filtered EEG trial x is standardized by applying the electrode-wise exponential moving standardization with a decay factor of 0.999 [29,31], according to the following equations:(1)$$\mu \left(t_{i}\right)=0.001x\left(t_{i}\right)+0.999\mu \left(t_{i-1}\right)$$
(2)$$\sigma^{2}\left(t_{i}\right)=0.001{\left(x\left(t_{i}\right)-\mu \left(t_{i}\right)\right)}^{2}+0.999\sigma^{2}\left(t_{i-1}\right)$$
(3)$$\tilde{x}\left(t_{i}\right)=\left(x\left(t_{i}\right)-\mu \left(t_{i}\right)\right)/\sigma \left(t_{i}\right)$$
which compute mean $\mu \left(t_{i}\right)$, variance $\sigma^{2}\left(t_{i}\right)$, and standardized $\tilde{x}\left(t_{i}\right)$ values on each electrode taken at sample $t_{i}$. The initial values $\mu \left(t_{0}\right)$ and $\sigma^{2}\left(t_{0}\right)$ are the mean and variance calculated over periods corresponding to the rest state preceding each trial. To rectify outliers, the EEG amplitudes of each trial are limited to $\mu \left(t_{i}\right)\pm 6\sigma \left(t_{i}\right)$. Finally, the trial crops strategy is employed for data augmentation. For both datasets, crops of 4 s every 8 ms in the interval from −0.5 to 4 s (cue onset at 0s) over all trials were extracted in our study.

### 2.3. Architecture

A shallow architecture that performs temporal and spatial convolutions is used here. The temporal convolutional layer with a 45 × 1 filter and 40 channels has input tensors of size 1000 × 22 × 1 and output tensors of size 478 × 22 × 40 when using dataset 2a, while input tensors of size 1000 × 3 × 1 and output tensors of size 478 × 3 × 40 are used for dataset 2b. Then, downsampling from 250 to 125 Hz by employing a stride of 2 is performed. The spatial convolutional layer is composed of 40 channels and a 1 × 22 filter when using dataset 2a and a 1 × 3 filter when using dataset 2b. After the temporal convolution and the spatial filter, a squaring nonlinearity, an average pooling layer with 45 × 1 sliding windows, a max-pooling layer with an 8 × 1 stride, and a logarithmic activation function are applied. These steps together are analogous to the trial log-variance computation, which is widely used in the Filter Bank Common Spatial Patterns (FBCSPs) [30,44]. The use of quadratic activation functions, or even higher-order polynomials, is not new in neural network research [45], and to the best of our knowledge, it was first used in BCI applications by Schirrmeister [31]. The classification layer is composed of a dense layer with Softmax activation function that receives a total of 2160 features. To avoid overfitting, batch normalization and dropout layers are used, and also the “MaxNorm” regularization is further applied in both convolution and dense layers. Moreover, the “Early Stopping” method and the decay of learning rate are also considered. The Adam optimizer [46] and the Categorical Cross-Entropy as a cost function are employed. As a result, the proposed architecture contains a total of 45,804 weights for dataset 2a and 11,082 weights for dataset 2b.

The neural network shown is totally deterministic and does not permit broader reasoning about uncertainty. To estimate the uncertainty, the Monte Carlo dropout described in the next section was used.

### 2.4. Monte Carlo Dropout

The dropout technique is commonly used to reduce the model complexity and also avoid overfitting [24]. A dropout layer multiplies the output of each neuron by a binary mask that is drawn following a Bernoulli distribution, randomly setting some neurons to zero in the neural network, during the training time. Then, the non-dropped trained neural network is used at test time. Gal and Ghahramani [23] demonstrated that dropout used at test time is an approximation of probabilistic Bayesian models in deep Gaussian processes. Monte Carlo dropout (MCD) quantifies the uncertainty of network outputs from its predictive distribution by sampling T new dropout masks for each forward pass. As a result, instead of one output model, T model outputs {**P***_t_*; 1 ≤ *t* ≤ *T*} for each input sample **x** are obtained. Then, the set $\left\{p_{t}\right\}$ can be interpreted as samples from the predictive distribution, which is useful to extract information regarding the prediction’s variability. This information is valuable for making decisions. In fact, quantifying the uncertainty of the model may allow uncertain inputs to be treated differently.

The main drawback of MCD is its computational complexity, which can be proportional to the number of forward passes T. As an alternative, the forward passes can run concurrently, resulting in constant running time. Moreover, if the dropout layers are located near the network output, as in the SCNN model (see Figure 2), the input of the first dropout layer can be saved in the first pass, to reuse it in the remaining passes, avoiding redundant computation [26]. Consequently, the computational complexity of MCD can be significantly reduced, enabling it for real-time applications.

The MCD model estimation can be computed as the average of T predictions:(4)$$p^{*}=\frac{1}{T}\sum_{t=1\;}^{T}p_{t}.$$

According to [23], T=50 is considered a safe choice to estimate the uncertainty, but this value must also be evaluated, considering the predictive performance of MCD. In our study employing the SCNN architecture (see Figure 2), the performance by applying MCD through different T samples was analyzed for each subject in both datasets 2a and 2b. Figure 3 shows the accuracy improvement (∆ ACC) from the baseline T=1. We observed that generally when T increases, the accuracy of SCNN-MCD improves, reaching an evident stabilization for values prior to T=50. For this reason, T=50 was adopted in SCNN-MCD.

The Monte Carlo dropout can be seen as a particular case of Deep Ensembles (training multiple similar networks and sampling predictions from each), which is another alternative to improve the performance of deep learning models and estimate uncertainty. A brief description of Deep Ensembles is presented in the next section.

### 2.5. Deep Ensemble Models

Deep ensembles have shown potential to improve the accuracy and out-of-distribution robustness as well as reduce the uncertainty of deep learning models [27,47]. The ensemble learning combines several models to obtain better generalization. Therefore, the disadvantages of using a single model can be tackled and masked by the strengths of other models. The averaged predictions are most useful when all models are statistically independent, having different hyperparameters, or being trained with different data.

Bagging [27,47], also known as bootstrap aggregating, is the type of ensemble technique in which a single training algorithm is used on different subsets over the same architecture. Bagging samples may be generated with/without replacement. Given a dataset, an ensemble predictor can be obtained by training the same architecture several times, where each training instance uses one bagging sample as training set. At prediction time, the same input is evaluated by each network, and the results are averaged. The main drawback of deep ensembles is their high computational cost and complexity for training and implementation.

To evaluate the generalization of DNN-based BCI systems, it is common to randomly partition the available data, defining a part for training, another set for validation, and the rest for testing. The pre-trained models from each partition can be added in an ensemble that corresponds to bagging without replacement [27,47]. This strategy is followed in our work for obtaining the ensemble model.

### 2.6. Uncertainty Analysis and Prediction Performance

The uncertainty in neural networks measures how reliable a model makes predictions. Several uncertainty measures can be used to quantify model uncertainty [48,49]. For a better understanding, we first present five well-known metrics, such as variation ratio $\left(V_{R}\right)$, predictive entropy (ℍ), mutual information (I), total variance $\left(V_{tot}\right)$, and margin of confidence (M). The next descriptions assume the aforementioned predictive distribution obtained from the stochastic forward passes.

Let C be the total number of classes for classification, and $p_{t}=\left(p_{1t},\;p_{2t},\;\cdots ,\;p_{Ct}\right)$; the model output for a forward pass, t, is 1≤t≤T. If the last layer of the model is softmax, the sum of all outputs is equal to 1. Let $p^{*}=\left(p_{1}^{*},\cdots ,\;p_{C}^{*}\right)$ be the average of the predictions $\left\{p_{t};1\leq t\leq T\right\}$.

Variation ratio $V_{R}$. This measures the dispersion or how spread the distribution is around the mode.
(5)$$V_{R}=1-\frac{f_{c^{*}}}{T}$$
where $f_{c^{*}}$ is the frequency of the mode $c^{*}$ of the discrete distribution $\left\{c_{t}\right\}$ and $\;c_{t}=arg\;max\left\{p_{1t},\;p_{2t},\;\cdots ,\;p_{Ct}\right\}$ is the predicted class in each stochastic forward pass.

Notice that $V_{R}\;\in \left[0,\;1/C\right]$, and it reaches its minimum and maximum values for $f_{c^{*}}$ closest to T and T/C, respectively.

Predictive Entropy ℍ. This metric captures the average of the amount of information contained in the predictive distribution. The predictive entropy attains its maximum value when all classes are predicted to have an equal uniform probability. In contrast, it obtains zero value as minimum when one class has a probability equal to 1, being 0 for all others (for instance, when the prediction is certain). The predictive entropy can be estimated as:(6)$$ℍ\approx \;\overset{C}{\underset{j=1}{\sum }}p_{j}^{*}\log_{2}(p_{j}^{*}).$$

The maximum value for ℍ is $\log_{2}C$. Therefore, it is not fixed for datasets with different numbers of classes. To facilitate the comparison across various datasets, we normalize the predictive entropy here as follows: ${ℍ}_{n}=ℍ/\log_{2}C$, ${ℍ}_{n}\in \left[0,1\right]$.

Mutual Information I. It measures the epistemic uncertainty by capturing the model’s confidence from its output.
(7)$$I\approx \;ℍ-\frac{1}{T}\overset{T}{\underset{t=1}{\sum }}\overset{C}{\underset{j=1}{\sum }}p_{jt}\log_{2}(p_{jt})$$

Total variance $V_{tot}$. It is the sum of variances obtained for each class:(8)$$V_{tot}=\frac{1}{T}\overset{C}{\underset{j=1}{\sum }}\overset{T}{\underset{t=1}{\sum }}{\left(p_{jt}-p_{j}^{*}\right)}^{2}.$$

Margin of Confidence M. The most intuitive form to measure uncertainty is analyzing the difference between two predictions of the highest confidence.

Let $c=argmax\;p_{j}^{*}$ be the predicted class through the MCD approach.

Then, for $d_{t}=p_{ct}-\underset{j\neq c}{max}p_{jt}$, we compute:(9)$$M=\frac{1}{T}\overset{T}{\underset{t=1}{\sum }}d_{t}$$
where M takes values close to zero for points toward high uncertainty, and it increases when the uncertainty decreases. We noted that M can be negative.

The prediction’s uncertainty can be intuitively expected to be correlated with the classification performance. For instance, Figure 4 shows the histograms of the normalized predictive entropy, for predictions classified correctly and incorrectly, when applying subject-specific classification on dataset 2a. We observed for almost all subjects that well-classified predictions were grouped predominantly toward low-entropy values, while the incorrect classified predictions were more clustered in regions of high entropy. A similar effect also occurred when applying the other uncertainty measure presented here. This indicates that the most uncertain predictions also tend to be incorrect. In areas of high uncertainty, the model can randomly classify patterns, and therefore, it is preferred to reject their associated inputs. The rejection decision can be carried out by using some uncertainty metrics, and preferably, it must be statistically inferred. Next, the more suitable uncertainty measures to achieve this purpose are determined.

As a novelty, a new approach based on the Bhattacharyya distance to compare the ability of several uncertainty measures for discriminating correct and incorrect classified predictions is proposed here in order to enhance the MI tasks recognition. The Bhattacharyya distance measures the similarity of two probability distributions *p* and *q* over the same domain *X*, and it can be calculated as
(10)$$D_{B}\left(p,\;q\right)=-\ln\left(\underset{x\in X}{\sum }\sqrt{p\left(x\right)q\left(x\right)}\right).$$

Table 1 shows the Bhattacharyya distance computed between histograms obtained from correct and incorrect classified predictions, using the aforementioned uncertainty measures. 

Notice that the margin of confidence M reached the highest Bhattacharyya distance on dataset 2a and the mean of both datasets, outperforming the other metrics. Thus, we used it in the classification process to reject those EEG trials that were less certain. The margin of confidence is a sample mean of 50 random values {$d_{t}$}; consequently, a normal distribution can be assumed for the random variable M. This allows fixing a threshold $\hat{M}\;$ on the values of M to split the predictions into certain $\left(M>\hat{M}\right)$ and uncertain $\left(M\leq \hat{M}\right)$. Notice that if the prediction is certain, the zero value must be outside the confidence interval of M, and therefore, M  must be necessarily greater than $\sigma_{d}z_{1-\frac{\alpha }{2}}/\sqrt{T}$, where $\sigma_{d}$ is the standard deviation of samples {$d_{t}$}, $z_{1-\frac{\alpha }{2}}=\Phi^{-1}\left(1-\frac{\alpha }{2}\right)$, Φ is the cumulative distribution function (CDF) of the standard normal distribution, and 1−α is the confidence level. Consequently, as threshold, the following equation can be used:(11)$$\hat{M}=\frac{\sigma_{d}z_{1-\frac{\alpha }{2}}}{\sqrt{T}}.$$

The certainty condition is satisfied if the mean M of the differences {$d_{t}$} is “very large” or if the standard deviation $\sigma_{d}$ is “very small”. As a result, this threshold scheme does not classify as uncertain those predictions in which the model is consistent $\left(\sigma_{d}\approx 0\right)$, even when M is close to zero. As a highlight, the proposed threshold does not require prior knowledge of the data, as it depends exclusively on the predictive distribution.

Finally, four subsets for predictions can be obtained by using the proposed method, which are incorrect–uncertain (iu), correct–uncertain (cu), correct–certain (cc), and incorrect–certain (ic) predictions.

Let $N_{iu},$ $N_{cu}$, $N_{cc}$, and $N_{ic}$ be the number of predictions in each subset, N be the total number of predictions, and $R_{c}$ be the certain ratio. This last ratio is the proportion of certain predictions with respect to the total number of predictions.

In any recognition system, the correct classification of certain predictions is desirable. Then, the correct-certain ratio $R_{cc}\;$ in Equation (12) [50] can be used to measure this expectation.
(12)$$R_{cc}\left(\hat{M}\right)=\frac{P\left(\mathrm{correct}\;\cap^{\;}\mathrm{certain}\right)}{P\left(\mathrm{certain}\right)}=\frac{N_{cc}}{N_{cc}+N_{ic}}$$

On the other hand, if the model makes an incorrect prediction, it is desirable to have high uncertainty, which can be measured by the incorrect–uncertain ratio $R_{iu}$ [50], as follows:(13)$$\;\;\;\;R_{iu}\left(\hat{M}\right)=\frac{P\left(\mathrm{uncertain}\;\cap^{\;}\;\mathrm{incorrect}\right)}{P\left(\mathrm{incorrect}\right)}=\frac{N_{iu}}{N_{iu}+N_{ic}}.$$

The overall accuracy of the uncertainty estimation can be measured through the Uncertainty Accuracy (UA) as:(14)$$UA\left(\hat{M}\right)=\frac{N_{cc}+N_{iu}}{N}=1-\frac{N_{ic}+N_{cu}}{N}$$
where $UA\left(\hat{M}\right)$ penalizes the incorrect–certain and correct–uncertain predictions, aiming to increase the reliability, effectivity, and feasibility of EEG MI-based recognition systems in practical applications. UA takes higher values for the best threshold values; thus, it can be further used to compare different thresholds.

### 2.7. Experiments

A first experiment to evaluate the accuracy performance by applying the SCNN-MCD approach was carried out. For the second experiment, a Monte Carlo dropout of an ensemble (E-SCNN-MCD) was executed in order to verify its feasibility to discriminate MI tasks. Finally, a third experiment to analyze the uncertainty of predictions during MI tasks classification was performed for both SCNN-MCD and E-SCNN-MCD approaches. All experiments were designed according to the conditions of the BCI competition IV to compare directly with recent works that also employed this dataset. For this, the training set (the first session from dataset 2a and the first three sessions from dataset 2b) was employed to calibrate the recognition system, while the testing set (the second session from dataset 2a and the last two sessions from dataset 2b) was used only for evaluation. A repeated holdout validation over the same testing set was carried out for both subject-specific and non-subject-specific classification strategies to evaluate the model generalization. For instance, the training set T  was split randomly into new sets $T_{1},\;V_{1}\;$, one for training ($T_{1})$ and the other for validation ($V_{1}$), repeating this random procedure 16 times ($T=T_{i}\cup^{\;}V_{i};1\leq i\leq 16)$. Once the model was trained for each $T_{i}$ and $V_{i}$, the average accuracy was calculated by using the testing set ℰ. Figure 5a,b show the strategy to select the training and testing sets for both subject-specific and non-subject-specific strategies, respectively. As a result, when applying the subject-specific classification, the training data are composed randomly of trials from the training set of the same subject, as shown in Figure 5a for Subject 5. In contrast, when using the non-subject-specific strategy, the training data are selected randomly from the training set of all subjects, as shown in Figure 5b for Subject 5.

An HPC using a Dell PowerEdge R720 server with four 2.40 GHz Intel Xeon processors and 48 GB RAM, NVIDIA GM107L Tesla M10 GPU with 32 GB memory was used to train and test the deep learning models by using the Python TensorFlow 2.3.0 framework.

## 3. Results and Discussion

### 3.1. Experiment #1: Monte Carlo Dropout to Improve MI Classification

Prior to the implementation of Monte Carlo dropout in the Shallow Convolutional Neural Network (SCNN) of Figure 2, several experiments were carried out on datasets 2a and 2b, using SCNN in [29] for both subject-specific and non-subject-specific classification. Once the model was trained, we then evaluated the Monte Carlo dropout accuracy on testing set ℰ.

For a subject-specific session to session classification (subject-specific system), the training set T from dataset 2a (first session) was composed of 288 trials per subject, while from dataset 2b, a total of 400 trials per subject (first three sessions) was used. The validation set $V_{i}$ was formed with 1/6 and 1/5 of the former training set T when using dataset 2a and dataset 2b, respectively. Table 2 and Table 3 show for our subject-specific system the averaged accuracy obtained on both dataset 2a and dataset 2b, respectively, as well as comparison with relevant state-of-the-art studies. Both tables highlight the best accuracies in bold, while the lowest accuracies of SCNN-MCD with respect to SCNN [29] are underlined.

Table 2 and Table 3 show that the SCNN-MCD method improved the mean ACC for both databases (around 2% for dataset 2a and 0.22% for dataset 2b) with respect to SCNN, surpassing the other analyzed methods except for CWT-SCNN on dataset 2b. SCNN-MCD reached the best results on subjects A03, A04, A05, A06, A07, and B07 compared with the state-of-the-art.

Interestingly, applying the SCNN-MCD approach for the subject-specific strategy did not improve the accuracy for some subjects (A09, B01, B02, B05, and B09), which suggests co-adaptation [24,51]. When dropout is applied at test time, the dropped neurons may degrade the model’s accuracy, especially if the neurons are relying too much on each other to make the prediction. The co-adaptation in the neural network is defined as the expected performance decrease when the dropout is applied at test time. In order to verify the co-adaptation scenario for these subjects, a Monte Carlo Dropout with T=1 was performed, taking into account 16 pre-trained weights for each subject. Figure 6 shows that for subjects A09, B01, B02, B05, and B09, the mean accuracy decreased for T=1 with respect to SCNN, indicating possibly a co-adaptation scenario.

For a non-subject-specific session to session classification (non-subject-specific system), as shown in Figure 5b, the training set from dataset 2a was composed of 2592 trials (288 trials per subject), while the training set from dataset 2b was composed of 3600 trials (400 trials per subject). Table 4 and Table 5 (for both datasets 2a and 2b, respectively) show the results obtained for non-subject-specific classification. Both tables highlight the best accuracies in bold, while the lowest accuracies of SCNN-MCD compared to [29] are underlined.

Table 4 and Table 5 show that the SCNN-MCD approach increased the mean ACC on both databases (2.5% for dataset 2a and 1.4% for dataset 2b), outperforming generally the SCNN approach. The SCNN-MCD improved its accuracy for most subjects, in comparison to SCNN, although it also decreased the ACC on some subjects (A06, A07, B03, and B05), indicating possibly co-adaptation [24,51]. The highest ACC improvement occurred on subjects A01 (8%), A02 (7%), and A05 (4.75%) from dataset 2a as well as on B01 (6.55%) from dataset 2b.

### 3.2. Experiment #2: Monte Carlo Dropout of an Ensemble to Improve MI Classification

Here, the Monte Carlo dropout of an ensemble of SCNN, termed as E-SCNN-MCD, is evaluated and compared with other methods. Then, from previous experiments, 16 trained models $M_{i}$ per subject were obtained. It allowed the testing of an ensemble with 16 models $\left\{M_{1},M_{2},\cdots ,M_{16}\right\}$. This ensemble can be seen as a particular case of bootstrap aggregating [27,47] in which bagging samples are performed with replacement. To accomplish this experiment, a Monte Carlo dropout was implemented in SCNN as aforementioned, where each prediction $p_{t}$, 1<t<50 of an ensemble was obtained by averaging the corresponding predictions of each model $M_{i}$, using dropout at test time.

The results obtained on dataset 2a are shown in Table 6 for both subject-specific and non-subject-specific classification. For both strategies, E-SCNN-MCD reached superior results compared to SCNN, with an improvement of 4.88% for subject-specific classification and 4.39% for non-subject-specific classification. With respect to SCNN-MCD, E-SCNN-MCD improved 3.03% for subject-specific classification and 1.89% for non-subject-specific classification. Similarly, Table 7 shows the results of E-SCNN-MCD on dataset 2b. For the subject-specific strategy, this approach improved 1.08% compared to SCNN and 0.86% with respect to SCNN-MCD. Furthermore, for non-subject-specific classification, E-SCNN-MCD increased the accuracy 2.87% and 1.43% compared to SCNN and SCNN-MCD, respectively. These results using ensemble are promising, as it reused trained models without another particular design.

### 3.3. Experiment #3: Uncertainty and Prediction Performance

This experiment was carried out to analyze the uncertainty of predictions and evaluate the certainty condition based on the proposed threshold (see Equation (11)), splitting the predictions into both certain and uncertain groups for both SCNN-MCD and E-SCNN-MCD approaches, enhancing the decision making. Similar to previous experiments, subject-specific and non-subject-specific classifications were considered. For this purpose, the confidence level (1−α) at 0.95 was selected. To assess the suitability of using the proposed threshold with this confidence level, different thresholds from 0.1 to 10 were tested, and their corresponding $R_{c},\;R_{cc},\;R_{iu}$, and UA values were also calculated. Figure 7 shows the uncertainty accuracy (UA) achieved for each threshold, using both subject-specific and non-subject-specific strategies. It is worth noting that UA values for thresholds corresponding to confidence levels of 0.95 and 0.99 are very close to the optimal value $UA^{*}$ for both strategies (see Figure 7a,b), which is remarkable. Particularly, the difference between $UA^{*}$ and UA (corresponding to α = 0.05) is less than 0.1%.

Table 8 shows $R_{c},\;\;R_{cc},\;R_{iu}$, and UA values, using α=0.05 for subject-specific classification on both datasets.

The SCNN-MCD approach provided different results of certainty in both datasets 2a and 2b, as shown in Table 8. Although for both datasets, the Uncertainty Accuracy (UA) reached approximately an optimum value in which around 10% of the predictions were labeled as uncertain on dataset 2a. The highest predictions rejection was observed on subjects A02 (18.38%) and A06 (19.69%). On dataset 2b, it only rejected 3% of its predictions, which was influenced greatly by subjects B02 (5.6%) and B03 (6.66%). Notice that for datasets 2a and 2b, around 21% and 7% of the misclassified predictions were labeled as uncertain, respectively.

A better idea—analyzing two uncertainty metrics—is given in Figure 8. It shows the bivariate distribution of the mutual information and the margin of confidence over predictions obtained on subjects A04 and B01, as well as the marginal distributions. The average of margin of confidence was similar on both subjects; however, we found a large variation in the mutual information. The average mutual information for subject B01 was four times lower than for subject A04. Although only these subjects have been considered here, it is worth mentioning that the average for both datasets 2a and 2b had similar behavior. Given that mutual information captures the model’s confidence in its output, it demonstrates that the SCNN-MCD approach provided more reliable outputs on dataset 2b over dataset 2a.

The achieved ACC for certain predictions $\left(R_{cc}\right)$ when using dataset 2a was 80.32% with an improvement of 2.89% with respect to SCNN-MCD, whereas for dataset 2b, it reached 78.31% with a marginal improvement of 0.59%.

For non-subject-specific classification, Table 9 shows the results achieved for both datasets, using confidence level  (1−α) at 0.95.

Table 9 shows that the SCNN-MCD approach also presented the highest uncertainty for dataset 2a, rejecting approximately 14% of its predictions, with a big influence from subjects A02 (18.75%), A05 (18.87%), and A06 (19.03%). For dataset 2b, it also rejected approximately 5% of its predictions, which was mainly influenced by subjects B02 (7.29%) and B03 (7.71%). For dataset 2a, more than 24% of misclassified predictions were considered as uncertain, and it was lower than 10% on dataset 2b. The accuracy achieved using certain predictions $\left(R_{cc}\right)$ of dataset 2a was 71.32% with an enhancement of 3.45% compared to SCNN, while it reached 77.63% with a marginal improvement of 0.86% for dataset 2b. As a highlight, the subject A03 in dataset 2a presented the best $R_{cc}$ (above 85%), whereas the subjects B04 and B08 in dataset 2b achieved $R_{cc}$ higher than 93%.

The previous analysis was extended to the ensemble model (E-SCNN-MCD), and the results are shown in Table 10 and Table 11 (for subject-specific and non-subject-specific classification, respectively), using both datasets and confidence level (1−α) at 0.95.

As expected, E-SCNN-MCD obtained predictions of highest certainty, being 97.89% and 99.05% on datasets 2a and 2b, respectively, when applying subject-specific classification. When comparing the results in Table 8 and Table 10, we observed that UA values improved by using the E-SCNN-MCD approach compared to SCNN-MCD. This improvement occurred despite the fact that there was a decrease in the number of uncertain incorrect predictions as is reflected in a decrease in $\;R_{iu}$. However, this decrease was compensated by a substantial increase in the number of correct certain predictions, which were expressed in the $R_{cc}$ indicator.

Figure 9 shows the bivariate distribution of predictions by applying both mutual information and margin of confidence for Subject A05, using the SCNN-MCD and E-SCNN-MCD approaches. The mutual information decreased using E-SCNN-MCD, compared with respect to SCNN-MCD, although it was without substantial changes for the margin of confidence. It is worth mentioning that this effect also occurred for all subjects in both datasets.

Similar behavior can be observed in Table 11 when applying non-subject-specific classification.

## 4. Conclusions

The advantages of applying the MCD method to enhance the performance of MI-based BCI schemes using deep neural network models have been proved in this study. Here, this approach was applied with the Shallow Convolutional Neural Network architecture, and an ensemble model, in order to validate its potential to improve subject-specific and non-subject specific MI classification and provide uncertainty estimation and consequently increase the reliability of BCIs. A threshold approach using uncertainty measures was also introduced here and applied on both SCNN-MCD and E-SCNN-MCD models to refuse automatically EEG trials that produce predictions of high uncertainty, obtaining lower error rates during MI classification. This proposed threshold does not require prior knowledge of the data, as it depends exclusively on the predictive distribution. In addition, it reaches its value near the expected optimum value for different MI datasets, using a confidence level at 0.95. Then, due to its statistical base, the selected threshold can be extended to other datasets. In future work, the proposed threshold including other uncertainty metrics can be explored to reject better EEG data that produce bad classified predictions. For clinical translation, this research has enormous potential due to the EEG variability, mainly in people with severe motor impairments, who increase the uncertainty of BCI predictions.

## Figures and Tables

**Figure 1 sensors-21-07241-f001:**
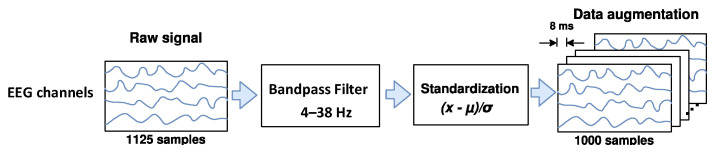
The raw EEG signals’ preprocessing and data augmentation.

**Figure 2 sensors-21-07241-f002:**
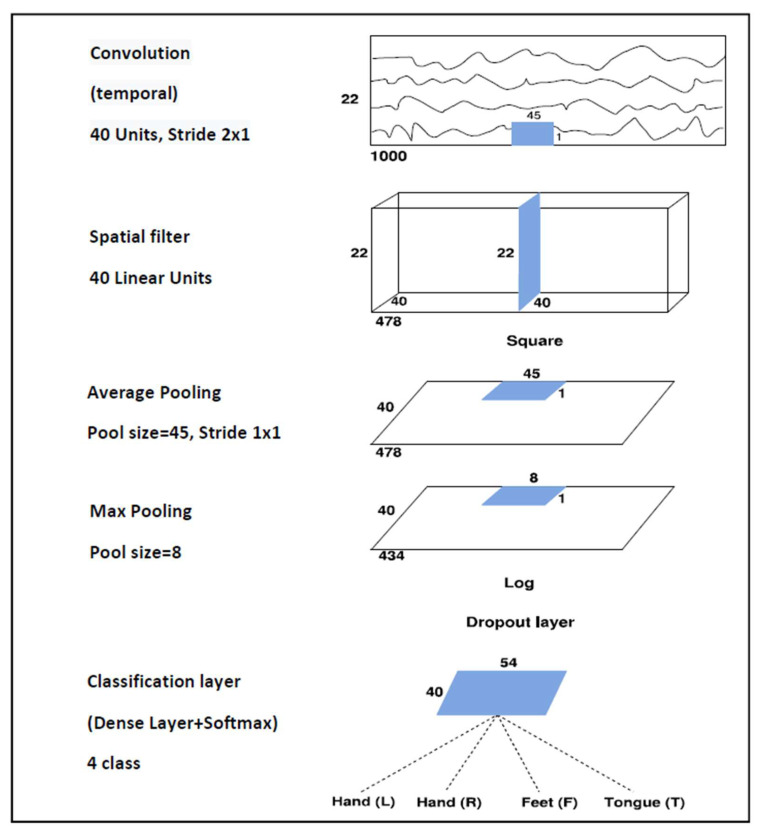
The Shallow Convolutional Network architecture proposed for the BCI Competition IV dataset 2a.

**Figure 3 sensors-21-07241-f003:**
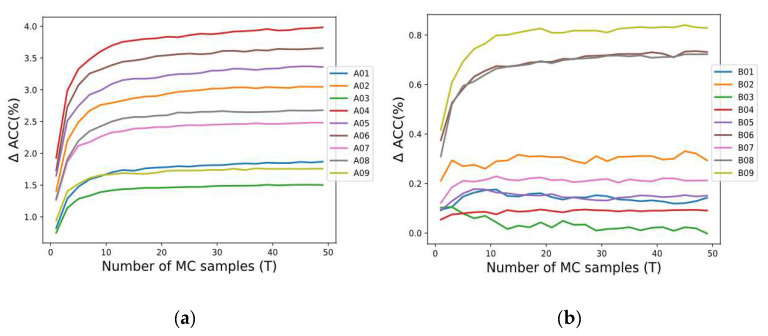
The performance using Monte Carlo dropout for several forward passes: (**a**) Subjects in dataset 2a; (**b**) Subjects in dataset 2b.

**Figure 4 sensors-21-07241-f004:**
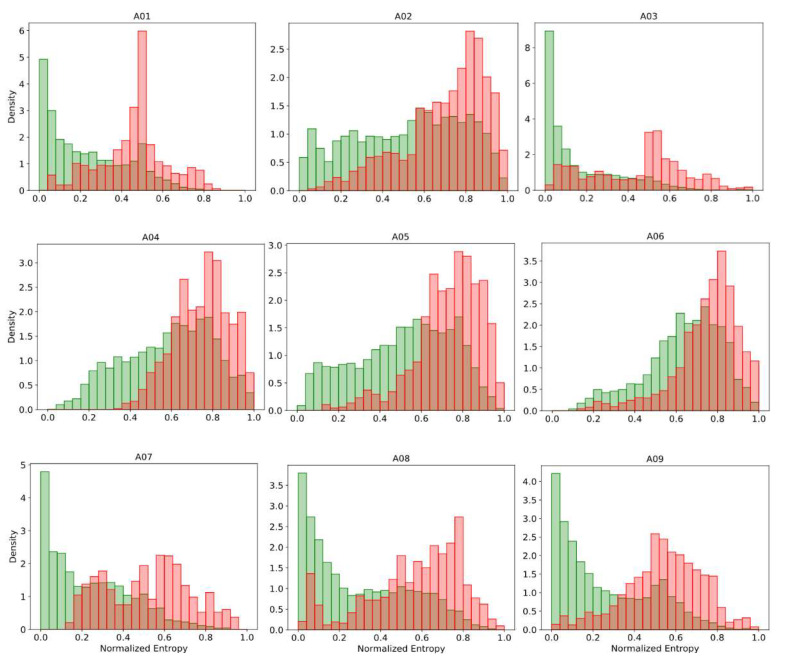
The histograms of predictive normalized entropy for correct (green) and incorrect (red) classified predictions on BCI Competition IV dataset 2a.

**Figure 5 sensors-21-07241-f005:**
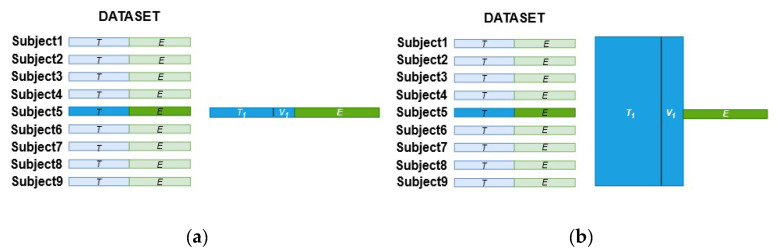
The training and testing sets selection for both strategies: (**a**) Subject-specific; (**b**) Non-subject-specific.

**Figure 6 sensors-21-07241-f006:**
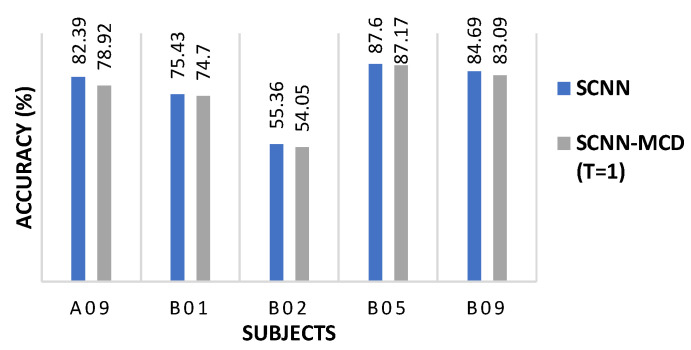
The SCNN-MCD performance with *T* = 1 compared to SCNN on selected subjects during subject-specific session to session classification.

**Figure 7 sensors-21-07241-f007:**
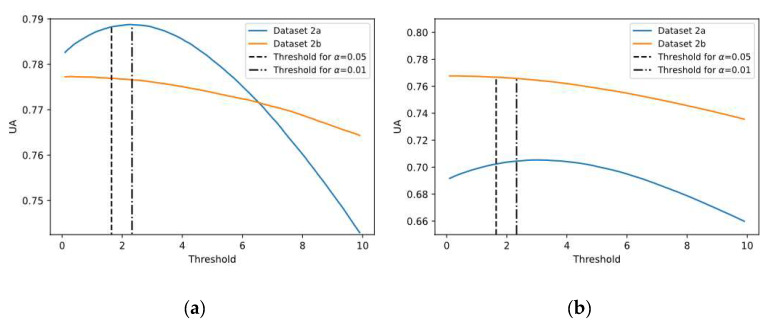
The mean Uncertainty Accuracy (UA) for different thresholds using datasets 2a and 2b: (**a**) Subject-specific strategy; (**b**) Non-subject-specific strategy.

**Figure 8 sensors-21-07241-f008:**
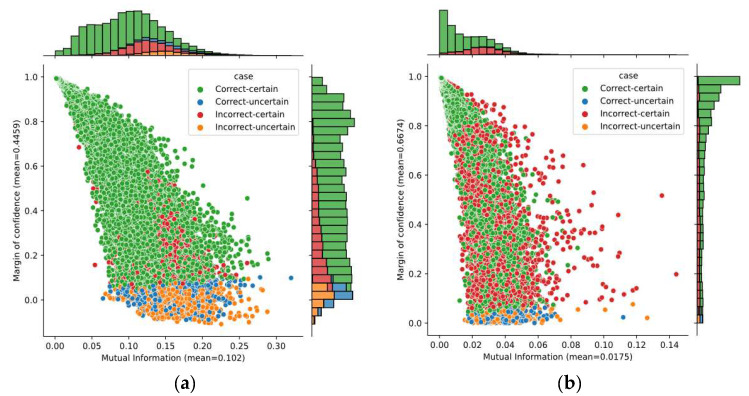
The bivariate distribution of predictions by using mutual information and margin of confidence: (**a**) Subject A04; (**b**) Subject B01.

**Figure 9 sensors-21-07241-f009:**
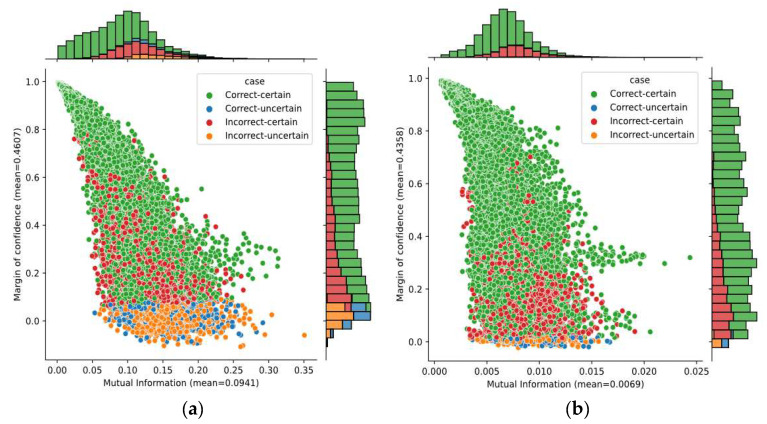
The bivariate distribution of predictions by using mutual information and margin of confidence for Subject A05: (**a**) SCNN-MCD approach; (**b**) E-SCNN-MCD approach.

**Table 1 sensors-21-07241-t001:** The Bhattacharyya distance between distributions of correct and incorrect classified predictions for different uncertainty measures. The best values are highlighted in bold.

**Dataset**	$V_{R}$	H	I	$V_{tot}$	M
2a	0.1500	0.1365	0.1331	0.1522	0.1589
2b	0.0862	0.1715	0.1638	0.1465	0.1657
Mean	0.1181	0.1540	0.1485	0.1494	0.1623

**Table 2 sensors-21-07241-t002:** The subject-specific session to session classification results (accuracy in percentage) on dataset 2a. The best accuracies are highlighted in bold; the lowest accuracies of SCNN-MCD with respect to SCNN are underlined.

Methods	A01	A02	A03	A04	A05	A06	A07	A08	A09	Mean
FBCSP [30]	76.00	56.28	80.88	61.07	54.85	45.48	82.98	81.63	70.45	67.70
SCN [31]	86.56	62.29	89.86	65.61	55.19	48.47	86.07	78.41	76.05	72.05
C2CM [32]	87.50	**65.28**	90.28	66.67	62.50	45.49	89.58	83.33	79.51	74.46
MCNN [34]	**90.21**	63.40	89.35	71.16	62.82	47.66	90.86	**83.72**	82.32	75.72
CCNN [34]	87.14	63.10	86.76	68.29	63.61	48.32	87.73	80.17	78.83	73.77
GLOBAL [33]	88.60	55.90	86.70	71.00	66.50	56.00	88.40	80.90	77.10	74.60
SCNN [29]	83.81	51.97	91.48	73.82	69.82	53.90	91.17	81.87	**82.39**	75.58
**SCNN-MCD**	85.27	57.11	**92.48**	**75.42**	**74.38**	**57.02**	**92.15**	82.36	80.68	**77.43**

**Table 3 sensors-21-07241-t003:** The subject-specific session to session classification results (accuracy in percentage) on dataset 2b. The best accuracies are highlighted in bold; the lowest accuracies of SCNN-MCD with respect to SCNN are underlined.

Methods	B01	B02	B03	B04	B05	B06	B07	B08	B09	Mean
CNN-SAE [35]	**78.10**	63.10	60.60	95.60	78.10	73.80	70.00	71.30	**85.00**	75.10
CWT-SCNN [36]	74.70	**81.30**	**68.30**	**96.30**	**92.50**	**86.90**	73.40	**91.60**	84.40	**83.20**
CAgross [37]	68.31	55.10	54.61	91.14	80.17	72.23	67.79	88.65	80.23	73.13
SCNN [29]	75.43	55.36	52.09	94.96	87.60	79.71	79.77	87.87	84.69	77.50
**SCNN-MCD**	74.84	54.34	52.44	95.84	87.32	82.04	**80.42**	88.34	83.92	77.72

**Table 4 sensors-21-07241-t004:** The non-subject-specific session to session classification results (accuracy in percentage) on dataset 2a. The best accuracies are highlighted in bold; the lowest accuracies of SCNN-MCD with respect to SCNN are underlined.

Methods	A01	A02	A03	A04	A05	A06	A07	A08	A09	Mean
SCN [31]	47.06	31.22	41.02	33.19	41.57	34.71	43.09	46.01	51.78	41.07
MCNN [34]	51.91	38.06	43.34	35.81	41.50	31.11	48.09	45.01	51.29	42.09
CCNN [34]	62.07	42.44	63.12	52.09	49.96	37.16	62.54	59.32	69.43	55.34
SCNN [29]	72.29	39.26	81.59	60.90	54.03	**51.58**	**74.70**	77.11	76.86	65.37
**SCNN-MCD**	**80.39**	**46.28**	**82.91**	**62.65**	**58.78**	49.76	71.51	**79.76**	**78.76**	**67.87**

**Table 5 sensors-21-07241-t005:** The non-subject-specific session to session classification results (accuracy in percentage) on dataset 2b. The best accuracies are highlighted in bold; the lowest accuracies of SCNN-MCD with respect to SCNN are underlined.

Methods	B01	B02	B03	B04	B05	B06	B07	B08	B09	Mean
SCNN [29]	66.94	55.66	**54.08**	93.78	**79.09**	80.81	74.18	90.05	83.42	75.33
**SCNN-MCD**	**73.49**	**56.94**	53.35	**94.42**	78.45	**81.28**	**76.16**	**92.73**	**84.15**	**76.77**

**Table 6 sensors-21-07241-t006:** The classification results (accuracy in percentage) by applying the E-SCNN-MCD model on dataset 2a.

E-SCNN-MCD	A01	A02	A03	A04	A05	A06	A07	A08	A09	Mean
Subject-specific	87.47	60.90	93.60	80.93	78.43	62.96	94.21	84.62	80.98	80.46
Non-subject-specific	80.97	47.37	83.87	64.89	62.05	52.15	75.25	80.86	80.44	69.76

**Table 7 sensors-21-07241-t007:** The classification results (accuracy in percentage) by applying the E-SCNN-MCD model on dataset 2b.

E-SCNN-MCD	B01	B02	B03	B04	B05	B06	B07	B08	B09	Mean
Subject-specific	74.92	54.30	53.33	96.22	88.02	84.09	80.59	89.37	86.36	78.58
Non-subject-specific	74.82	58.67	54.35	94.96	80.91	84.11	77.11	93.89	84.98	78.20

**Table 8 sensors-21-07241-t008:** The uncertainty metrics (%) obtained by using the SCNN-MCD scheme for subject-specific session to session classification.

Dataset 2a						Dataset 2b				
Subjects	UA *	Rc	Rcc	Riu	UA	UA *	Rc	Rcc	Riu	UA
1	85.33	95.71	86.74	13.87	85.06	74.87	96.98	75.63	6.04	74.86
2	63.04	81.62	61.27	26.31	61.29	54.39	94.40	54.58	6.09	54.31
3	92.52	97.83	93.42	14.42	92.47	52.68	93.34	52.63	7.04	52.47
4	76.92	87.30	79.97	28.86	76.91	95.85	99.47	96.08	6.34	95.84
5	75.67	89.26	78.02	23.42	75.64	87.35	98.48	87.90	6.04	87.33
6	62.59	80.31	61.44	27.95	61.36	82.04	97.41	82.84	6.92	81.94
7	92.24	96.75	93.48	19.60	91.98	80.43	97.50	81.18	6.28	80.38
8	83.03	92.66	85.25	22.50	82.96	88.35	97.87	89.12	8.66	88.23
9	81.88	93.59	83.33	19.25	81.70	83.93	97.30	84.82	8.18	83.85
Mean	78.88	90.56	80.32	21.80	78.82	77.73	96.97	78.31	6.84	77.69

* indicates optimal value.

**Table 9 sensors-21-07241-t009:** The uncertainty metrics (%) obtained by using the SCNN-MCD scheme for non-subject-specific session to session classification.

Dataset 2a						Dataset 2b				
Subject	UA *	Rc	Rcc	Riu	UA	UA *	Rc	Rcc	Riu	UA
1	80.70	91.59	83.25	21.77	80.52	73.50	94.94	74.72	9.45	73.44
2	60.08	81.25	49.05	22.95	52.18	56.99	92.71	57.49	8.48	56.95
3	83.54	92.67	85.82	23.13	83.48	53.32	92.29	53.55	8.10	53.20
4	65.58	83.68	66.62	25.21	65.16	94.42	99.18	94.77	7.11	94.39
5	63.98	81.13	63.34	27.83	62.86	78.45	95.31	79.74	10.36	78.23
6	57.91	80.97	52.71	23.78	54.62	81.28	95.39	82.71	11.89	81.12
7	72.52	85.12	75.75	27.55	72.32	76.16	96.41	77.09	7.35	76.08
8	81.19	90.85	83.51	25.95	81.12	92.73	98.31	93.43	11.10	92.65
9	79.84	92.01	81.84	21.33	79.83	84.17	96.91	85.17	9.27	84.00
Mean	70.54	86.59	71.32	24.39	70.23	76.77	95.72	77.63	9.24	76.68

* indicates optimal value.

**Table 10 sensors-21-07241-t010:** The uncertainty metrics (%) obtained by using the E-SCNN-MCD scheme for subject-specific session to session classification.

Dataset 2a						Dataset 2b				
Subjects	UA *	Rc	Rcc	Riu	UA	UA *	Rc	Rcc	Riu	UA
1	87.68	98.80	87.96	5.06	87.54	75.19	98.86	75.25	2.43	75.00
2	65.28	95.34	62.25	7.95	62.46	54.61	98.29	54.38	1.89	54.31
3	93.75	99.62	93.80	3.61	93.68	53.32	98.16	53.29	1.76	53.13
4	81.33	97.42	81.94	7.75	81.30	96.25	99.80	96.29	2.23	96.18
5	78.97	97.76	79.23	5.85	78.72	88.17	99.57	88.23	2.15	88.11
6	64.86	95.61	63.79	6.55	63.41	84.16	99.07	84.39	2.81	84.05
7	94.32	99.37	94.51	5.71	94.25	81.04	99.09	80.87	2.33	80.59
8	85.29	98.50	85.33	6.02	84.98	90.00	99.45	89.62	2.85	89.42
9	84.03	98.60	81.54	4.32	81.22	86.40	99.18	86.66	2.98	86.36
Mean	81.36	97.89	81.15	5.87	80.84	78.64	99.05	78.78	2.38	78.57

* indicates optimal value.

**Table 11 sensors-21-07241-t011:** The uncertainty metrics (%) obtained by using the E-SCNN-MCD scheme for non-subject-specific session to session classification.

Dataset 2a						Dataset 2b				
Subjects	UA *	Rc	Rcc	Riu	UA	UA *	Rc	Rcc	Riu	UA
1	82.34	98.04	81.71	5.74	81.20	75.54	98.74	75.16	2.62	74.87
2	52.08	95.74	48.02	5.45	48.84	58.68	97.62	58.81	2.69	58.52
3	85.15	98.84	84.45	4.75	84.24	54.55	98.01	54.39	2.09	54.26
4	69.04	96.16	66.11	7.19	66.09	95.11	99.76	95.05	2.16	94.93
5	65.94	96.67	62.85	5.37	62.80	80.91	98.81	81.25	2.94	80.84
6	56.72	96.33	52.68	4.73	53.01	84.21	98.90	84.52	3.62	84.17
7	75.33	97.59	75.91	4.99	75.31	77.30	98.98	77.43	2.38	77.18
8	81.57	98.14	81.71	6.19	81.38	93.91	99.67	94.04	2.84	93.90
9	82.09	98.53	81.07	4.62	80.78	85.32	99.24	85.25	2.58	84.99
Mean	72.16	97.34	70.50	5.45	70.41	78.27	98.86	78.43	2.66	78.19

* indicates optimal value.

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
