# Peer review of "Monte Carlo Dropout for Uncertainty Estimation and Motor Imagery Classification"

_sensors, 2021, doi:10.3390/s21217241_

Round 1
Reviewer 1 Report
A research article (manuscript ID: sensors-1414979) entitled “Monte Carlo Dropout for Uncertainty Estimation and Motor Imagery Classification” by an international collaborative research team from the USA, Cuba, Brazil, and Colombia was submitted to the MDPI Sensors Journal.
This paper has 19 pages including 9 figures, 11 tables, and 44 references. In their investigations, the authors provided a method to estimate the model uncertainty and also introduce a threshold method to discriminate predictions of high uncertainty. In their work, the authors used the BCI Competition IV Databases 2a and 2b to evaluate the proposed methods, obtaining promising results for both subject-specific and nonsubject-specific strategies. The Motor Imagery (MI)-based Brain-Computer Interfaces (BCIs) have been widely used as an alternative communication channel to patients with severe neuromotor disabilities, achieving high classification accuracy by machine learning techniques. This work can be interesting for researchers as well as for graduate and postgraduate students studying this subject.
First of all, it is necessary to state that all the table (but tables 6 and 7) should have the horizontal line at the bottom.
Also, the references should be in the MDPI format. There are also many references without page numbers. These references rifer to arXiv but not to a Journal directly. For instance, there are references [6, 9, 10, 22, 23, 24, 25, 26, 39, 40, 43].
The reviewer has looked through the paper and found that this presentation requires some revision.
For instance, the following correction of the English language can be done in Abstract:
1) line 18: “MI-based BCIs but these techniques” instead of “MI-based BCIs, but these techniques”;
2) line 27: “nonsubject” instead of “non-subject”;
So, the authors should polish the English language of their presentation.
Please, use always “The” in any figure title after each figure number.
The same for the tables.
“Figure 1. The preprocessing” instead of “Figure 1. Preprocessing”;
“Figure 2. The Shallow Convolutional Network architecture” instead of “Figure 2. Shallow Convolutional Network architecture”;
“Table 1. The Bhattacharyya distance” instead of “Table 1. Bhattacharyya distance”;
The English language must be polished. The corresponding figures and tables must be improved. All the references must be in the MDPI format.. The paper requires a major revision.
Author Response
Dear Editor,
The authors would like to thank the reviewer for the very helpful comments of the editor and the reviewers in order to improve the quality of our paper. We understand that the reviewer's work has taken a large amount of time, and we are especially grateful for this. All of the comments and suggestions were taken into account for the revised version of our research report. We are uploading: (a) our point-by-point response to the comments (below) (response to reviewers), (b) an updated manuscript with yellow highlighting indicating changes, and (c) a clean updated manuscript without highlights (PDF main document).
Best regards,
José Núñez et al.
Reviewer 2 Report
This paper employs MC-dropout for quantifying model uncertainties of deep neural networks in EEG MI-based BCI systems. The authors conducted experiments on the cases of a single neural network and ensemble model for both subject-specific and non-subject-specific tasks. Furthermore, this manuscript presents the improved reliability of the MI-based BCI system by rejecting predictions with high uncertainties.
Overall, the topic is interesting, and this paper presents diverse experimetal results for demonstrating the effectiveness of the uncertainty information. However, the presentation of the current manuscript has room for improvement, and I believe that this manuscript is adequate for publication after major revision. The reviewer's comments are as follows.
[1] (Band-pass filtering) In the preprocessing step, band-pass filtering is applied for the frequencies ranging from 4 Hz to 38 Hz. It is required to explain the cut-off frequencies utilized in the band-pass filtering from medical points of view.
[2] (Standardization) Details for the process of standardization are insufficient. Although the authors refer to the previous studies [28, 31], the method utilized in the proposed method has to be explained in the main text. I recommend adding explanations for the electro-wise exponential movement standardization. Furthermore, the authors have to clarify whether the standardization was conducted for each subject or it was conducted for the entire data samples.
[3] (Network architecture) The authors have to clarify the definition of Square and logarithmic activation functions. Because these activation functions are rarely utilized in popular neural network architectures, it is beneficial to add the reasons for selecting these nonlinearities.
[4] (MC-Dropout) The authors have to revise the sentences in lines 168~171 to clarify the process of uncertainty quantification.
[5] (Equations) Several terms in (3)~(6) are not defined in the corresponding texts.
[6] (Figure 4) The resolution of texts in Figure 4 has to be improved.
[7] (Tables) Several numbers are incorrectly bolded (e.g. A06 and A07 in Table 4, B03 and B05 in Table 5)
[8] There are several typos, grammatical errors, and incorrectly defined abbreviations including:
- Deep learning neural network (DNN) (line 32)
- thecnique in line 152
- turns-off in line 153
- grammatical error in line 164
- The abbreviation of SCNN in line 168 is not defined.
- the dataset 2a y 2b in line 259
- Duplicated definitions for MCD in line 486
Author Response

(The authors gave the same response as above.)

Round 2
Reviewer 1 Report
A research article (revised manuscript ID: sensors-1414979) entitled “Monte Carlo Dropout for Uncertainty Estimation and Motor Imagery Classification” by an international collaborative research team from the USA, Cuba, Brazil, and Colombia was submitted to the MDPI Sensors Journal.
This revised paper has 20 pages including 9 figures, 11 tables, and 55 references. In their investigations, the authors provided a method to estimate the model uncertainty and also introduce a threshold method to discriminate predictions of high uncertainty. In their work, the authors used the BCI Competition IV Databases 2a and 2b to evaluate the proposed methods, obtaining promising results for both subject-specific and nonsubject-specific strategies. The Motor Imagery (MI)-based Brain-Computer Interfaces (BCIs) have been widely used as an alternative communication channel to patients with severe neuromotor disabilities, achieving high classification accuracy by machine learning techniques. This work can be interesting for researchers as well as for graduate and postgraduate students studying this subject.
The referee see that almost the half of the paper was correcte-improved, i.e. it is in the yellow color. However, there are still many references without page numbers. These references rifer to arXiv but not to a Journal directly.
Author Response
Dear Editor,
The authors deeply appreciate the voluntary contribution of the editor and each reviewer in the form of valuable comments that helped to improve our manuscript. Every effort has been taken by the authors to revise the manuscript according to the editor’s and reviewers’ comments.
We are uploading: (a) our point-by-point response to the comment (below) (response to reviewers), and (b) an updated manuscript with yellow highlighting indicating changes.
Best regards,
Authors
This revised paper has 20 pages including 9 figures, 11 tables, and 55 references. In their investigations, the authors provided a method to estimate the model uncertainty and also introduce a threshold method to discriminate predictions of high uncertainty. In their work, the authors used the BCI Competition IV Databases 2a and 2b to evaluate the proposed methods, obtaining promising results for both subject-specific and nonsubject-specific strategies. The Motor Imagery (MI)-based Brain-Computer Interfaces (BCIs) have been widely used as an alternative communication channel to patients with severe neuromotor disabilities, achieving high classification accuracy by machine learning techniques. This work can be interesting for researchers as well as for graduate and postgraduate students studying this subject.
Reviewer#1: The referee see that almost the half of the paper was correcte-improved, i.e. it is in the yellow color. However, there are still many references without page numbers. These references rifer to arXiv but not to a Journal directly.
Author response:
We are very grateful for your recommendation. We have edited and revised our manuscript as per the suggestion. We apologize for the mistakes in our earlier manuscript concerning references. We have fixed these mistakes according to the MDPI format. Because we did not find all the information of some preprints we have eliminated these bibliographical references and replaced them with new ones. Also, the numbering of some references has been updated in the text following the order of citation. Hence, we have inserted the bibliographical references according to Reviewer 1's recommendations.
Reviewer#2: The authors conducted major revisions according to the reviewer's comments, and I believe that this manuscript satisfies requirements for publication.
Author response:
All authors involved in our study really appreciate your kind comments. We also thank you for your valuable comments along the all revision process to improve the quality of the paper.
Reviewer 2 Report
The authors conducted major revisions according to the reviewer's comments, and I believe that this manuscript satisfies requirements for publication.
Author Response
Dear Editor,
The authors deeply appreciate the voluntary contribution of the editor and each reviewer in the form of valuable comments that helped to improve our manuscript. Every effort has been taken by the authors to revise the manuscript according to the editor’s and reviewers’ comments.
We are uploading: (a) our point-by-point response to the comment (below) (response to reviewers), and (b) an updated manuscript with yellow highlighting indicating changes.
Best regards,
Authors
Reviewer#1: The authors conducted major revisions according to the reviewer's comments, and I believe that this manuscript satisfies requirements for publication.
Author response:
We are very grateful for your recommendation. We have edited and revised our manuscript as per the suggestion. We apologize for the mistakes in our earlier manuscript concerning references. We have fixed these mistakes according to the MDPI format. Because we did not find all the information of some preprints we have eliminated these bibliographical references and replaced them with new ones. Also, the numbering of some references has been updated in the text following the order of citation. Hence, we have inserted the bibliographical references according to Reviewer 1's recommendations.